# OpenReview forum: "Advancing LLM Reasoning Generalists with Preference Trees"
_ICLR.cc/2025/Conference — ICLR 2025 Poster_

### Official Review · Reviewer_rVp1 · 2024-11-03

**Soundness:** 2
**Presentation:** 3
**Contribution:** 3
**Rating:** 6
**Confidence:** 3

**Summary:**

This paper presents EURUS, a new collection of large language models (LLMs) and a reward model (RM) designed to enhance reasoning capabilities. The authors develop ULTRAINTERACT, a dataset designed for complex reasoning tasks with 12 datasets spanning math, coding, and logical reasoning problems. ULTRAINTERACT employs preference trees, which pair each instruction with reasoning chains, interaction trajectories with feedback, and pairwise responses for preference learning.

The authors use ULTRAINTERACT to fine-tune several open-source LLMs, including Mistral-7B, Llama-3, and Mixtral-8x22B. They show that EURUS models achieve top performance on multiple reasoning benchmarks, including LeetCode and TheoremQA. EURUS-7B and LLAMA-3-EURUS-8B even surpass baselines 5 times their size, while EURUX-8X22B outperforms GPT-3.5 Turbo on 12 test sets.

They also create a reward model, EURUS-RM-7B, that excels on several reward modeling benchmarks and introduce a new reward modeling objective that merges the Bradley-Terry objective with an additional term to directly adjust the reward of chosen and rejected action

**Strengths:**

1. The paper introduces a novel dataset, ULTRAINTERACT, designed for complex reasoning tasks. It comprises instructions paired with preference trees, featuring reasoning chains, multi-turn interaction trajectories with feedback, and pairwise positive and negative responses. ULTRAINTERACT emphasizes complex and diverse reasoning patterns, encouraging models to break down problems into sub-problems and use tools to solve them. This dataset is a valuable contribution and can be useful for future research on LLM reasoning.

2. The proposed EURUS models achieve state-of-the-art performance on several reasoning benchmarks, demonstrating the effectiveness of ULTRAINTERACT and the proposed training methods. Notably, the smaller EURUS models outperform much larger baselines, showcasing their efficiency.

3. The paper provides valuable insights into preference learning for reasoning tasks. The analysis of reward patterns during training leads to a new reward modeling objective that improves performance, particularly on challenging problems. The authors highlight the importance of the absolute value of rewards in preference learning for reasoning, as opposed to just focusing on relative differences as in general conversation settings.

**Weaknesses:**

1. While the authors acknowledge the use of proprietary GPT models in data synthesis, they do not thoroughly analyze the limitations of relying on these models. It would be helpful to discuss the potential biases introduced by GPT models and explore alternative approaches for data generation that rely solely on open-source models. Though, it's worth noting that they attempt to address this by creating ULTRAINTERACT-v2 using only open-source models, which shows promising results.

2. In the paper, a few preference learning algorithms, since the preference pairs are collected in the ULTRAINTERACT, not running RL with the data seems like a big miss.

**Questions:**

1. In line 90-91, the statement is unclear to me. In 'a higher final reward often indicates a better reasoning capability', whose reasoning capability? Can you elaborate a bit more?

2. About the result remove in Table 3 due to data contamination. For some of the model has data contamination issue, the table suggests the TheoryQA is leaked, what about the rest dataset? If the rest doesn't has data contamination issue, should the result be compared? Without TheoryQA number, OpenChat seems like still a strong candidate.

---

> ### Author Response · Authors · 2024-11-21
> **Response to Reviewer rVp1**
>
> > Q1. While the authors acknowledge the use of proprietary GPT models in data synthesis, they do not thoroughly analyze the limitations of relying on these models. It would be helpful to discuss the potential biases introduced by GPT models and explore alternative approaches for data generation that rely solely on open-source models. Though, it's worth noting that they attempt to address this by creating ULTRAINTERACT-v2 using only open-source models, which shows promising results.
>
> A1: Thanks for your suggestion. Major concerns of using proprietary models lie in the non-permissive license and the financial cost of calling APIs. Using open-source models can help address both issues.We acknowledge that certain biases might exist. However, for reasoning tasks where there are absolute right and wrong, we believe such biases are less severe than general conversations where preferences can be subjective and ambiguous. Rather, by mixing multiple open-source models, we can introduce more diverse patterns and intervene potential spurious correlations, which may help mitigate such biases.
>
> > Q2. In the paper, a few preference learning algorithms, since the preference pairs are collected in the ULTRAINTERACT, not running RL with the data seems like a big miss.
>
> A2: We considered only preference learning because of its simplicity. It’s known that PPO models are hard to train, so it will introduce many confounders to track the effect of our data. However, following your suggestion, we are implementing PPO experiments in this discussion period but it takes time. We will update results later once it is finished.
>
> > Q3. In line 90-91, the statement is unclear to me. In 'a higher final reward often indicates a better reasoning capability', whose reasoning capability? Can you elaborate a bit more?
>
> A3: Sorry for the confusion. By better reasoning capability we mean higher benchmark results as indicated in Table 3. We have clarified the description.
>
> > Q4. About the result remove in Table 3 due to data contamination. For some of the model has data contamination issue, the table suggests the TheoryQA is leaked, what about the rest dataset? If the rest doesn't has data contamination issue, should the result be compared? Without TheoryQA number, OpenChat seems like still a strong candidate.
>
> A4: Thanks for pointing this out. We can only confirm the contamination of TheoremQA. Even if we simply remove the number of TheoremQA, Eurus-7B-SFT (48.9), Eurus-7B-KTO (51.4), Eurus-7B-NCA (50.6) still outperform OpenChat (48.7).

---

### Official Review · Reviewer_EWRx · 2024-11-04

**Soundness:** 3
**Presentation:** 3
**Contribution:** 4
**Rating:** 8
**Confidence:** 4

**Summary:**

This paper has several contributions. First, it builds a dataset on reasoning tasks that contain both correct and wrong steps. Second, it proposed a modified loss function for training a reward model that is better suited for reasoning tasks. Lastly, it trains a set of LLMs using the proposed dataset that have competitive performance on reasoning tasks.

**Strengths:**

- The paper is advancing open science by making the training data and model checkpoints public. Given the significant improvements in reasoning tasks, it is likely that these assets will be helpful to other researchers.
- The paper also proposes a new way of training reward models that is better suited to reasoning tasks. In addition, the training datasets have multi-step attempts that contain mistakes and tool usage, which is unlike other preference datasets.
- The experimental section is detailed and provides many interesting results, such as comparing three different preference optimization methods. There are many ablations provided, and evaluations are done on many tasks, which makes the results more convincing.

**Weaknesses:**

- The heavy reliance on GPT responses makes me feel like this is more of distilling GPT. Also, it is not clear what are the usage limitations that will arise from using a proprietary model like GPT4. As shown in tab7, this was crucial for obtaining good performance.
- The problem of the likelihood of chosen responses going down in reasoning is a known issue and studied prior work [1], which is not cited in the paper (the related work is quite short)
- The term “multi-turn action” was confusing. It seems that all the tasks require only a single correct response. None of the tasks is truly multi-turn where the model has to do multiple actions. From reading the paper, it seems the term “multi-turn” is used to describe a process where a model can try again if it makes a mistake. Actually, it is not clear how this process works, especially when training the model and evaluating it. Also, the dataset contains observations and judgements, but are they also used when training the actor? What about the python executions? There is very little detail on how the agent is trained on these and evaluated.
- As mentioned in the previous point, there are certain steps that are not well explained. See the questions for examples. Given that the motivation is to advance open-source LLMs, I think it is important to describe the process of training in more details.

**Questions:**

- Is the reward model used in training the actor model?
- L148 “the actor model first decomposes the input problem into several problems” How is this done?
- L181 “we adopt more diverse reasoning patterns” How exactly is this done?
- Is python the only tool used?
- Typo in L263 reward notation
- What is "prompt level loose score" in L282
- I think the tables have too many numbers in them (tab3 has at least a hundred) and not sure if anyone will look at all of them. Instead, average scores can be put there and the detailed table can move to the appendix. This is only a suggestion though.
- Which GPT-4 model is used? I think there are multiple versions.
- How is the reward model performance compared to ArmoRM?
- How is GPT-4 used as a reward model in tab4?
- Why does self-consistency drop in fig1 left?
- How is MCTS decoding done exactly in sec5.2?

---

> ### Author Response · Authors · 2024-11-21
> **Response to Reviewer EWRx**
>
> Thanks for your positive feedback!
>
> > Q1. The heavy reliance on GPT responses makes me feel like this is more of distilling GPT. Also, it is not clear what are the usage limitations that will arise from using a proprietary model like GPT4. As shown in tab7, this was crucial for obtaining good performance.
>
> A1: Thanks for your comments but we think there could be some misunderstanding. It is Table 6 that aims to ablate the effect of GPT rather than Table 7. In Table 6, we show that models trained on UltraInteract V2, the version constructed using only open-source models, can outperform the models trained on V1, the GPT generated version. This gives us a clue that we can construct high-quality data without GPT. We suppose the “open-source only” row in Table 7 distracts you, which indeed means training SFT models with only the open-source data (UltraChat, ShareGPT, and OpenOrca) without UltraInteract. Experiments in table 7 demonstrate that the superior performance of our SFT model is credited to our carefully generated data and its rationales. We have updated Table 7 and refer to the setup as “existing data only” for clarification.
>
> Regarding the usage limitations of proprietary models, major concern may lie in the non-permissive license and the financial cost of calling APIs. Using open-source models can help address both issues.
>
> > Q2. The problem of the likelihood of chosen responses going down in reasoning is a known issue and studied prior work [1], which is not cited in the paper (the related work is quite short)
>
> A2: Thanks for pointing it out. Our related work is short because we are already short of space, but we will add the reference you list. It seems the paper title is missing, would you mind specifying it?
>
> > Q3. The term “multi-turn action” was confusing. It seems that all the tasks require only a single correct response. None of the tasks is truly multi-turn where the model has to do multiple actions. From reading the paper, it seems the term “multi-turn” is used to describe a process where a model can try again if it makes a mistake. Actually, it is not clear how this process works, especially when training the model and evaluating it. Also, the dataset contains observations and judgements, but are they also used when training the actor? What about the python executions? There is very little detail on how the agent is trained on these and evaluated.
>
> A3: Thanks for your comments and sorry for the confusion. We directly follow the setup in MINT [1], which measures the multi-turn correction ability of LLMs. All questions in our dataset can be answered within a single turn, but LLMs can make mistakes so they may require another turn of action to correct its previous answers. We acknowledge that there are some agentic tasks that may intrinsically require interruption of actions to wait for observations and feedback, where LLMs have to act multiple times before finally answering for one time. However, according to CodeAct [2], as long as we use code as actions, which is exactly what we did, we can design all control flows in a single code block (i.e., an action) and compress it into a single turn. All intermediate results can be automatically saved in the code variables and be forwarded for further processing. Therefore, we believe this setup can already cover all multi-turn scenarios that require multiple actions.
>
> During inference, LLMs generate <execute></execute> when they need to write code. Upon finishing generating the whole response, we extract all code between the tags and send it to a sandbox to execute, and append the returned outputs to the end of the model response as observations. However, when training actors, even though observations and feedback are also provided in the history, they will be masked and only the model generated tokens will be optimized.
>
> [1] MINT: Evaluating LLMs in Multi-turn Interaction with Tools and Language Feedback. Wang et al. ICLR 2024.
>
> [2] Executable Code Actions Elicit Better LLM Agents. Wang et al. ICML 2024.
>
> > Q4. As mentioned in the previous point, there are certain steps that are not well explained. See the questions for examples. Given that the motivation is to advance open-source LLMs, I think it is important to describe the process of training in more details.
>
> A4: Thanks for your suggestions. We have provided more details in the Appendix.
>
> > Q5. Is the reward model used in training the actor model?
>
> A5: No. During dataset construction, the actor model is fixed; After obtaining the data and subsequent reward model, we can utilize the reward model train models with PPO, but in this paper we directly conducted preference learning algorithms due to their simplicity.
>
> > Q6. L148 “the actor model first decomposes the input problem into several problems” How is this done?
>
> A6: Our setup is highly consistent with MINT [1]. We prompt the model to reason before writing code as actions, and think step by step to solve the problem.

---

> ### Author Response · Authors · 2024-11-21
> **Response to Reviewer EWRx (2)**
>
> > Q7. L181 “we adopt more diverse reasoning patterns” How exactly is this done?
>
> A7: We have already described the process in Line 150-152: To promote solution diversity, the actor model randomly samples one reasoning schema in the form of either CoT (Wei et al., 2022) or modularization programming (Qian et al., 2023; Yuan et al., 2023).
>
> > Q8. Is python the only tool used?
>
> A8: Yes. However,  the Python code interpreter in our case is equipped with external libraries/tools like wiki search. As explored in many recent works [1, 2, 3], the Python interpreter provides a general-purpose and broadly applicable environment for various tasks and applications. For example, a Python interpreter enables LLMs to not only defer calculations but also call tools to search for information or perform complex real-world agent tasks, such as sending emails or online shopping [1, 4].
>
> [1] Executable Code Actions Elicit Better LLM Agents.
>
> [2] FireAct: Toward Language Agent Fine-tuning.
>
> [3] Taskweaver: A Code-First Agent Framework.
>
> [4] Tool Learning with Foundation Models.
>
> > Q9. Typo in L263 reward notation
>
> A9: Thanks for pointing this out. We have corrected the issue.
>
> > Q10. What is "prompt level loose score" in L282
>
> A10: IFEval provides four metrics in their codebase: prompt-level strict score, prompt-level loose score, instruction-level strict score, instruction-level loose score. We directly follow the evaluation setup in ??? Detailed discussion on this metric is beyond the scope of this work.
>
> > Q11. I think the tables have too many numbers in them (tab3 has at least a hundred) and not sure if anyone will look at all of them. Instead, average scores can be put there and the detailed table can move to the appendix. This is only a suggestion though.
>
> A11: Thanks for your suggestion. We will highlight the average scores, but we think providing a detailed breakdown of the model performance may help readers understand what is going on so we tend to keep them.
>
> > Q12. Which GPT-4 model is used? I think there are multiple versions.
>
> A12: We always used the latest version of GPT-4. Specifically, we started with gpt-4-0613 to provide feedback, and then switched to gpt-4-1106-preview and gpt-4-0125-preview.
>
> > Q13. How is the reward model performance compared to ArmoRM?
>
> A13: On RewardBench, ArmoRM achieves a higher score than EurusRM (89.0 and 82.4). However, It is worth noting that ArmoRM uses a stronger base model (LLaMA-3-8B) and a much larger training dataset. Specifically, we use 803K pairs while they use 587.4K for the multi-objective reward modeling and another 1004.4K for the gating layer training. Therefore, it may not be a fair baseline of our work.
>
> > Q14. How is GPT-4 used as a reward model in tab4?
>
> A14: We directly adopt results from [1]. According to their paper, they use LLM-as-a-judge to rank two responses using the following prompts:
>
> ```
> Instruction: ${instruction}
>
>
>
> Input: ${input}
>
>
>
> Candidate A: ${candidate1}
>
>
>
> Candidate B: ${candidate2}
>
>
>
> Given the instruction and input above, please compare the two candidates. You only have 4 choices to output:
>
>
>
> If you think A is better, please output: 1. A is better
>
>
> If you think B is better, please output: 2. B is better
>
>
>
> If you think both are good enough correctly give the answer, please output: 3. Same good
>
>
>
> If you think both are bad and do not follow the instruction, please output: 4. Same bad
>
>
>
> Do not output anything else except the 4 choices above.
> Output your choice below:
> ```
>
> [1] LLM-BLENDER: Ensembling Large Language Models with Pairwise Ranking and Generative Fusion. Jiang et al. ACL 2023.
>
> > Q15. Why does self-consistency drop in fig1 left?
>
> A15: Increasing N may include some low-quality data into consideration and dilute the proportion of correct answers, which may distract self-consistency and finally lead to a different major voted answer.
>
> > Q16. How is MCTS decoding done exactly in sec5.2?
>
> A16: We implement the vanilla MCTS decoding setup using LLMReasoner [1].
>
> [1] LLM Reasoners: New Evaluation, Library, and Analysis of Step-by-Step Reasoning with Large Language Models. Hao et al. COLM 2024.

---

> > ### Comment · Reviewer_EWRx · 2024-12-03
> >
> > Thanks for the detailed response. I will maintain my score. Sorry for the missing reference.
> >
> > [1] Iterative reasoning preference optimization, Pang et al, 2024

---

### Official Review · Reviewer_rmjd · 2024-11-07

**Soundness:** 2
**Presentation:** 3
**Contribution:** 3
**Rating:** 6
**Confidence:** 3

**Summary:**

The authors emphasize the performance gap between open-source LLMs and the most advanced models, particularly in reasoning capabilities. They attribute this gap to two primary factors: (1) the lack of high-quality datasets and (2) the under-exploration of preference learning techniques. To address this gap, the authors introduce a novel dataset, ULTRAINTERACT, which features a multi-turn, tree-structured format designed to enhance reasoning abilities. Additionally, they offer new insights into preference algorithms and reward modeling. They argue that effective reward modeling should consider not only the margin between rewards but also the absolute value of the reward itself.  Based on this insight, they propose a new reward model that combines two loss functions, L_{BT} and L_{DR}, demonstrating superior performance compared to existing models.

**Strengths:**

Authors use a new method to synthesize a dataset for SFT and preference learning, which could potentially enhance model's reasoning abilities. The intuition behind the synthesis method is straightforward and easy to be understood. I think the dataset is cool and it could be a potential approach for model to learn how to improve the response. Plus, the insights on preference learning algorithm is interesting.

**Weaknesses:**

1). I agree that providing trajectories to guide model improvements is a potential approach. However, during the training process, I believe that the vertical improvement information, sequential refinement across turns, may not be effectively learned. This is because current preference algorithms primarily focus on horizontal comparisons, assessing responses within the same turn.

2). The reasons behind the better performance of EURES are hard to track and some studies will be necessary if authors want to claim that the proposed dataset is the reason. Because the baselines has different scales and training method, for example, their training dataset could have different size and their preference algorithm could be different, etc.. Plus if EURES can beat some larger model, the claim that the dataset is better will be more convincing.

3). There may be some factors contributing to the value differences observed in reward modeling, especially given the varying formulations of alignment methods. It would be valuable for the authors to offer insights into the potential reasons for these differences in the value of rewards.

**Questions:**

If the model is unable to effectively learn from vertical improvements, then it raises the question of why we want to synthesize the dataset with tree structure and why we are providing trajectories to the model.

---

> ### Author Response · Authors · 2024-11-21
> **Response to Reviewer rmjd (1)**
>
> Thank you for your constructive comments. We will try our best to address your concerns.
>
> > Q1: I agree that providing trajectories to guide model improvements is a potential approach. However, during the training process, I believe that the vertical improvement information, sequential refinement across turns, may not be effectively learned. This is because current preference algorithms primarily focus on horizontal comparisons, assessing responses within the same turn.
>
> A1: Thanks for your comments. We try to address your concern by emphasizing some empirical results.
>
> 1. To further demonstrate the effectiveness of the tree structure, we trained Llama-3-Eurus-8B-SFT on single-turn pairwise data, namely decomposing a multi-turn tree into multiple single turn pairs. Results can be found here. We find that compared to training on single-turn pairs, training on multi-turn trees enjoys huge benefits on multi-turn interaction ability and slightly improves the overall performance.
>
> | Model | Coding |     |     | Math |     |     |     |     | Reasoning | Ins-Following | Multi-Turn |     | Avg. |
> | --- | --- | --- | --- | --- | --- | --- | --- | --- | --- | --- | --- | --- | --- |
> |     | HumanEval | MBPP | LeetCode | GSMPLUS | MATH | TheoremQA | SVAMP | ASDiv | BBH (CoT) | IFEval | Code | Math |     |
> | Llama-3-Eurus-8B-SFT | 51.2 | 57.9 | 17.2 | 50.7 | 32.0 | 21.3 | 82.2 | 83.7 | 72.4 | 47.1 | 18.4 | 24.5 | 46.6 |
> | \+ KTO | 51.8 | 58.1 | 15.6 | 54.8 | 34.2 | 24.9 | 80.1 | 86.7 | 71.7 | 50.6 | 26.5 | 37.4 | 49.4 |
> | \+ KTO (single-turn) | 53.7 | 59.1 | 14.4 | 54.8 | 30.7 | 23.1 | 77.8 | 86.2 | 72.1 | 49.9 | 22.8 | 33.0 | 48.1 |
> | \+ NCA | 50.6 | 60.4 | 15.6 | 55.2 | 34.8 | 25.4 | 79.9 | 87.5 | 71.7 | 56.2 | 21.3 | 36.3 | 49.6 |
> | \+ NCA (single-turn) | 53.7 | 55.9 | 16.1 | 55.4 | 30.5 | 25.4 | 79.3 | 87.5 | 72.2 | 54.2 | 17.7 | 35.5 | 48.6 |
>
> 2. We mainly explored preference learning in this work. However, even if preference learning cannot effectively learn from the sequential refinement, this dataset may also facilitate other algorithms that target improving the refinement ability. For example, our data offers an opportunity to implement other algorithms such as SELF-CORRECTION [1], which trains a model to map an incorrect response to a correct one. One can implement this by adopting our incorrect response in previous turns and the refined correct response in the following turns as the training data pairs.
>
> [1] Generating Sequences by Learning to Self-Correct. Welleck et al. 2022.

---

> > ### Comment · Reviewer_rmjd · 2024-11-21
> >
> > I really appreciate for providing additional experiments to address my concerns. I might have misunderstood some parts, so I would like to ask what is 'decomposing a multi-turn tree into multiple single turn pairs'. Based on my understanding, even though the dataset follows a tree structure, when the data is fed into the preference algorithm for training, it should inherently be multiple single turn pairs. For example, if we use the tree in Figure 2 (Right) to train the model, it's inherent to be three pairs. Therefore, I would like to ask how this experiment was conducted and how the tree structure was preserved during training. Thanks again for authors' patience and thoughtful responses.

---

> > > ### Author Response · Authors · 2024-11-21
> > > **Respond to follow-up comments of Reviewer rmjd**
> > >
> > > Dear Reviewer,
> > >
> > > Thanks for your prompt response!
> > >
> > > By 'decomposing a multi-turn tree into multiple single turn pairs', we mean training models on the preference pairs without interaction history.
> > >
> > > More specifically, for experiments in our paper, we adopt different strategies to utilize interaction history. For SFT, as stated in line 244-245, "We find it yields better performance to discard interaction history and train only on correct leaf nodes in each tree.", i.e. only training on (instruction, single-turn response). However, for preference learning, "Differently from SFT, here we include all multi-turn trajectory pairs in our ULTRAINTERACT", i.e., given a tree of five turns, we will have five pairs with the depth of each being 1, 2, ..., 5. The pair at the later turn can oberve the full interactions between previous responses and environment/critique which aims to rectify previous wrong nodes. Namely, we train on (instruction, chosen response at turn 1, rejected response at turn 1),  (instruction + rejected response at turn 1 + observation + critique, chosen response at turn 2, rejected response at turn 2), etc. We hope that LLMs can learn from the interaction history so that they can rectify incorrect answers based on feedback during inference. We presented one case of the single-turn SFT and one case of multi-turn preference learning in Table 15 and 16 in Appendix G.2 respectively.
> > >
> > > In the additional experiments, we split a tree of five turns into five single-turn examples, namely (instruction, chosen response at turn 1, rejected response at turn 1), (instruction, chosen response at turn 2, rejected response at turn 2), etc. No interaction history is presented, which is akin to our setup of SFT. Intuitively, this omits the "vertical information" in our trees and thus will lead to performance drop in benchmark results on MINT. Our additional results have confirmed this intuition, therefore demonstrating that LLMs can effectively learn the "sequential refinement across turns" from our preference trees.
> > >
> > > If you have any further questions, please let us know and we will do our best to address your concerns. Thanks!

---

> > > > ### Comment · Reviewer_rmjd · 2024-11-25
> > > >
> > > > Thank you for patiently addressing my questions and providing the additional experiments. Your responses have clarified some of my concerns, and I will increase my score. I believe this dataset will be valuable for future research.

---

> ### Author Response · Authors · 2024-11-21
> **Response to Reviewer rmjd (2)**
>
> > Q2. The reasons behind the better performance of EURES are hard to track and some studies will be necessary if authors want to claim that the proposed dataset is the reason. Because the baselines has different scales and training method, for example, their training dataset could have different size and their preference algorithm could be different, etc.. Plus if EURES can beat some larger model, the claim that the dataset is better will be more convincing.
>
> A2: Thanks for your comments. However, we argue that despite the different model and dataset sizes, Eurus models actually beat baselines with larger backbone size, larger datasets, or more advanced algorithms, which should make our claims even more convincing.
>
> Firstly, we highlight that the size of our models and baselines are comparable. Most baseline models in ~7B category are Mistral-7B based, the same as our Eurus-7B. The exceptions are Magicoder-S-DS-6.7B and OpenCI-DS-6.7B, which are trained on DeepSeek-Coder-6.7B, a stronger base model on reasoning and especially on coding. However, our Eurus-7B and Llama-3-Eurus-8B outperforms all of them and even surpasses much larger ~40B models (Mixtral-8x7B-Instruct and Deepseek-Coder-33B-Ins).
>
> Secondly, most of the baselines did not open their dataset, so compared to those, our datasets and mixtures are transparent, demonstrating a huge advantage. Compared to the remaining models with open datasets, we trained Eurus-7B-SFT with 399K data in total, and for preference learning we used 560K pairwise data. However, baselines consume more data than us in terms of SFT, and do not open their recipe for preference learning:
>
> | **Model** | **SFT Data Size** | **Preference Learning Data Size** |
> | --- | --- | --- |
> | CodeLLaMA-70B-Instruct | Non-transparent | Non-transparent |
> | DeepSeek-LM-67B-Chat | 1.5M | Non-transparent |
> | QWen1.5-72B-Chat | Non-transparent | Non-transparent |
> | OpenCI-CL-70B | 68K Sample, 192K Turn | -   |
> | OpenMath-CL-70B | 1.8M | -   |
> | WizardLM-2-8x22B | Non-transparent | Non-transparent |
> | Mixtral-8x22B-Instruct-v0.1 | Non-transparent | Non-transparent |
> | Ours | 399K | 560K |
>
> Also, since our data mixture for SFT consists of UltraInteract and existing data, we conducted an ablation study in Section 6.2 in which we trained models either only on our data or only on existing open-source data. Results in Table 7 can firmly support the claim that the performance boost on reasoning is due to our proposed dataset.
>
> Lastly, it’s not clear what methods are used to train baseline models, but we may suppose that all models have at least gone through SFT as common practice, and any further operations may be intended to push the limit of their SFT models. Therefore, comparing our SFT models to baselines should be a fair setup to them but may not be fair to us. Nevertheless, our SFT models can already outperform baselines.
> Therefore, we consider that our comparisons are convincing and we can safely claim that our dataset is the major contributor of the superior performance.
>
> > Q3. There may be some factors contributing to the value differences observed in reward modeling, especially given the varying formulations of alignment methods. It would be valuable for the authors to offer insights into the potential reasons for these differences in the value of rewards.
>
> A3. The major difference shown in Figure 6 is that, the rewards of chosen data and margins increase regardless of $\mathcal{L}_{\text{DR}}$, but the rewards of rejected data decrease to be negative with regularization. This may be attributed to the nature of $\mathcal{L}_{\text{BT}}$, which only optimizes the relative margin between rewards if chosen data and rejected data, while not explicitly forcing the rewards of chosen data to be positive and that of rejected data to be negative. Therefore, the absolute value of rewards are not guaranteed.
>
> > Q4. If the model is unable to effectively learn from vertical improvements, then it raises the question of why we want to synthesize the dataset with tree structure and why we are providing trajectories to the model.
>
> A4. Please see response to weakness 2.

---

### Official Review · Reviewer_xgCx · 2024-11-12

**Soundness:** 3
**Presentation:** 3
**Contribution:** 3
**Rating:** 6
**Confidence:** 3

**Summary:**

The authors explore improving large language model reasoning through the curation of high quality training data for that reasoning.
This data (UltraInteract) consists in preference trees, with nodes splitting on correct/incorrect responses; critique and refinement of rejected responses; and uses different reasoning schemas/ actor models to increase training data diversity
The actor used to generate these trajectories is GPT3.5 Turbo, with GPT4 being used as a critique model with access to an interpreter/tools.

The authors then use this dataset (alongside others) to finetune 3 language models using the following process:
1. SFT over the correct actions
2. Preference learning over correct vs incorrect actions using off the shelf preference learning algorithms

Additionally the authors also use this to derive a reward model:
3. Train a reward model, adding in terms for the difference in absolute rewards to the normal Bradley Terrey reward model.

In my view the key contributions of this paper are:
* introduction and analysis of preference-tree based instruction following data, which is scalable and effective
* introduction of improved objectives for training reward models

**Strengths:**

1. With regards to soundness, I feel that the necessary experiments have been run to validate the majority of claims, especially where those claims are with regards to methodological contributions. The authors have also taken pains to remove contaminated data from their work in order to make comparisons fair and meaningful, including when reporting others' work.
2. The presented language models have strong performance, and the data and reward models are in and of themselves useful contributions to the research community, removing some of the limitations of scale and quality from prior works creating preference datasets and reward models
3. The investigation surrounding the flaws of existing preference learning models is an original contribution.
4. In my view the largest contribution is the rather detailed study of creating their ultra-instruct dataset albeit moreso as an engineering challenge.
5. The experiments are run against meaningful baselines: models of similar scale, trained on similar data in similar ways.

**Weaknesses:**

1. As a minor point the spelling and grammar could be improved; for instance "Is proprietary models" (line 470) should be "Are proprietary models", and more generally things like "Perference Learning" (line 247). More substantially some of the references point to the wrong sections (e.g. the reference to section 5 (replaced with 6) (line 255) -- in this case harming readability (hence my review of the presentation...)
2. I feel that the modification to the reward model could be better motivated in section 3, for instance by referencing other works that maximise a similar margin loss. At the least it should be explicitly linked to the discussion in section 4.2 that actually seems to motivate it. This might be aided to seperating out the reward modelling section from the finetuning section? Since it seems to follow on more logically from the finetuning investigations
3. Section 6.1 doesn't really address the section title properly. While the performance itself does suggest that just training on open source data is sufficient (ignoring the instruction following benchmark); the body of the section just talks about mixing in this additional V2 data, and the ensuing performance gains. It would suffice to add a brief comment at the end of line 483 explaining the results of finetuning just on V2
4. As a general comment I feel that this work feels like three distinct pieces of work rather than a single cohesive one. I.e. the proposal of a new training dataset; a set of models finetuned on this dataset alongside others; and more separetely a reward model trained on a combination of dataset including the one proposed here. One way of mitigating this would be to focus on the contribution of the dataset to the reward modelling phase (using the data from the ablation studies).
5. Section 2. is a little bit confusing and could be rephrased to make it a little but clearer that it is all just an example.

**Questions:**

1. Did you conduct any comparative investigations over general conversational preference learning using your reward modelling objective? This would help to verify your intuition that this method is effective due to the unique features of reasoning tasks
2. Would it be possible to use the Eurus reward model for PPO-based alignment? How would this perform in comparison to the existing finetuning methods

---

> ### Author Response · Authors · 2024-11-21
> **Response to Reviewer xgCx**
>
> Thank you for your positive feedback and diligent efforts in reviewing our paper. We appreciate your comments and would like to share our responses.
>
>
> > Q1: As a minor point the spelling and grammar could be improved; for instance "Is proprietary models" (line 470) should be "Are proprietary models", and more generally things like "Perference Learning" (line 247). More substantially some of the references point to the wrong sections (e.g. the reference to section 5 (replaced with 6) (line 255) -- in this case harming readability (hence my review of the presentation...)
>
> A1: Thanks for pointing out these typos. We have fixed them.
>
>
> > Q2: I feel that the modification to the reward model could be better motivated in section 3, for instance by referencing other works that maximise a similar margin loss. At the least it should be explicitly linked to the discussion in section 4.2 that actually seems to motivate it. This might be aided to seperating out the reward modelling section from the finetuning section? Since it seems to follow on more logically from the finetuning investigations
>
> A2: Thanks for your comment and sorry that we mistakenly linked to section 6 in section 3, instead of referring to section 4.2. We have corrected the reference and provided more explanations to motivate the objective.
>
>
> > Q3: Section 6.1 doesn't really address the section title properly. While the performance itself does suggest that just training on open source data is sufficient (ignoring the instruction following benchmark); the body of the section just talks about mixing in this additional V2 data, and the ensuing performance gains. It would suffice to add a brief comment at the end of line 483 explaining the results of finetuning just on V2
>
> A3: Thanks for your suggestion. We have added a description on results of training with merely V2: Compared to V1, training on V2 improves model performance on both SFT and preference learning stage, and particularly, the Llama-3-Eurus-8B-KTO (V2) successfully surpasses the official Llama-3-8b-Instruct model, which it previously failed to.
>
>
> > Q4: As a general comment I feel that this work feels like three distinct pieces of work rather than a single cohesive one. I.e. the proposal of a new training dataset; a set of models finetuned on this dataset alongside others; and more separetely a reward model trained on a combination of dataset including the one proposed here. One way of mitigating this would be to focus on the contribution of the dataset to the reward modelling phase (using the data from the ablation studies).
>
> A4: Thanks for your comments. Our ultimate goal is to build strong open-source reasoning generalists, which involves a lot of aspects. We aimed to make this work solid in execution so we compile all the artifacts, results and insights together. However, following your suggestion, we have emphasized the contribution of data in this recipe and highlighted our ablation results which can support this claim. To make the flow more coherent, we add a quick road map to the end of introduction:
>
> We compiled this work by first synthesizing both SFT and preference datasets to improve the reasoning ability of open-source models (Section 2). We examined the effectiveness of our datasets by training both policy and reward models (Section 3). We evaluated the performance of policy models in Section 4, during which we observed a correlation between reward patterns and benchmark performances. Next, we then evaluated our reward models and validated that our insights on the reward-performance correlation can be converted into gains in model training (Section 5). Finally, we ablate some factors in our dataset construction in Section 6.
>
>
> > Q5: Section 2. is a little bit confusing and could be rephrased to make it a little but clearer that it is all just an example.
>
> A5: Thanks for your suggestion. We have emphasized this in the updated version.

---

> ### Author Response · Authors · 2024-11-21
> **Response to Reviewer xgCx (2/2)**
>
> > Q6: Did you conduct any comparative investigations over general conversational preference learning using your reward modelling objective? This would help to verify your intuition that this method is effective due to the unique features of reasoning tasks
>
> A6: Thanks for your suggestion. We conducted an additional experiment on UltraFeedback (one pair per instruction) with $\mathcal{L}_{BT}$ and $\mathcal{L}_{DR}+\mathcal{L}_{BT}$ respectively. Results in the following table show that $\mathcal{L}_{DR}$ does not help improve reward model performance on general chat data, which may indicate that the absolute value of rewards are not that important as on reasoning tasks. This aligns with our intuition to only apply $\mathcal{L}_{DS}$ to UltraInteract examples.
>
> | Loss | Chat | Chat Hard | Reasoning | Safety |
> | --- | --- | --- | --- | --- |
> | $\mathcal{L}_{BT}$ | 94.5 | 44.1 | 56.5 | 52.9 |
> | $\mathcal{L}_{DR} + \mathcal{L}_{BT}$ | 92.8 | 36.05 | 45.6 | 43.0 |
>
> > Q7: Would it be possible to use the Eurus reward model for PPO-based alignment? How would this perform in comparison to the existing finetuning methods
>
> A7: We considered only preference learning because of its simplicity. It’s known that PPO models are hard to train, so it will introduce many confounders to track the effect of our data. However, following your suggestion, we are implementing PPO experiments in this discussion period but it takes time. We will update results later once it is finished.

---

> > ### Comment · Reviewer_xgCx · 2024-11-22
> >
> > Don't worry too much about implementing the PPO experiments, I appreciate the difficulty of running those experiments.
> > Thanks for updating the paper, it feels much more readable to me now, although there are still a few mistakes I'm sure youll find in the editing process
> >
> > As a quick question, did you have any ablation results of data mixtures used in preference learning? It would be nice to see the effect of using ultrainteract vs ultrafeedback here in particular, though I appreciate that we can expect this to be similar to the results on reward modelling so not necessary. But regardless would help to validate importance of the dataset, especially since the existing ablations are done on the SFT data which only uses the final responses from the preference trees if I am correct?

---

> > > ### Author Response · Authors · 2024-11-23
> > > **Reponse to follow-up comments of Reviewer xgCx**
> > >
> > > Dear Reviewer,
> > >
> > > Thanks for your response! Following your suggestion, we present the ablation study on prefernce data mixture as follows:
> > >
> > > | Model                                | Coding | Math  | BBH   | IFEval | Multi-Turn | Avg.  |
> > > | ------------------------------------ | ------ | ----- | ----- | ------ | ---------- | ----- |
> > > | Llama-3-Eurus-8B-SFT                 | 42.11  | 53.97 | 72.40 | 47.10  | 21.45      | 46.60 |
> > > | +KTO (UltraFeedback + UltraInteract) | 41.84  | 56.14 | 71.70 | 50.60  | 31.92      | 49.40 |
> > > | +KTO (Only UltraFeedback)            | 44.60  | 56.26 | 72.00 | 50.10  | 24.03      | 48.80 |
> > > | +KTO (Only UltraInteract)            | 40.70  | 55.86 | 71.70 | 50.60  | 34.49      | 49.40 |
> > >
> > > | Model                                 | MT-Bench |
> > > | ------------------------------------ | -------- |
> > > | Llama-3-Eurus-8B-SFT                 | 6.8      |
> > > | +KTO (UltraFeedback + UltraInteract) | 7.3      |
> > > | +KTO (Only UltraFeedback)            | 7.5      |
> > > | +KTO (Only UltraInteract)            | 7.2      |
> > >
> > > From the results, we see that training only on UltraInteract leads to higher overall reasoning performances. Looking deeper, these improvements are mainly credited to the multi-turn interaction ability, which demonstrates the superiority of the tree structure of our data. However, we also observe a lower MT-Bench score compared to training solely on UltraFeedback. Nevertheless, this can be mitigated without hurting reasoning performances by mixing these two datasets together, which indicates that our data is compatible with other datasets, consistent with our conclusions on reward modeling.

---

> > > > ### Comment · Reviewer_xgCx · 2024-11-25
> > > >
> > > > Great thanks for running those additional ablation studies, I think together with th studies you performed below on training on single-turn vs multi-turn interactions these make the contribution of the ultra-interact preference tree more convincing. Would it be possible to reflect this in the paper a bit more, for instance by adding these additional ablation studies to the appendix and referencing these in the claims made in 327-329. This would help to emphasise the contribution of ultra-interact as a multi-turn interaction dataset and improve the soundness of the claims made in your evaluation section in my view. I understand the space limitations however.

---

> > > > > ### Author Response · Authors · 2024-11-25
> > > > >
> > > > > Dear Reviewer,
> > > > >
> > > > > Thanks for your response! Following your suggestion, we revised the paper and supplemented the results in Appendix H. We also referenced them in line 329.
> > > > >
> > > > > If you have any further questions, please let us know and we will do our best to address your concerns. Thanks!

---

### Meta-Review · Area_Chair_J15Y · 2024-12-18

**Metareview:**

This paper presents a suite of large language models that achieve state-of-the-art performance in reasoning tasks. Key contributions include the introduction of a novel dataset featuring multi-turn preference trees designed for reasoning, and a reward modeling objective tailored to reasoning tasks. The paper demonstrates competitive performance across a variety of benchmarks, with EURUS models outperforming larger baselines and even proprietary GPT-3.5 Turbo in reasoning.

The reviewers agree that the contribution in this paper, despite being largely empirical, is valuable. Since  open model and data is a major contribution of the proposed work, the authors should consider disclose as much details as possible of their experimental process upon acceptance of the paper.

**Additional Comments On Reviewer Discussion:**

The reviewers raised various concerns including motivation of the new reward model design, the lack of cohesiveness of the three distinct pieces of work, heavy reliance on distilling from GPT, and writing clarities. These concerns are mostly addressed by author response.

---

### Decision · Program_Chairs · 2025-01-22

Accept (Poster)